# The Role of Tissue Transglutaminase in Cancer Cell Initiation, Survival and Progression

**DOI:** 10.3390/medsci7020019

**Published:** 2019-01-25

**Authors:** Claudio Tabolacci, Angelo De Martino, Carlo Mischiati, Giordana Feriotto, Simone Beninati

**Affiliations:** 1Department of Medicine, Campus Bio-Medico University, Via Alvaro del Portillo 21, IT-00128 Rome, Italy; 2Department of Biology, University of Rome “Tor Vergata”, Via della Ricerca Scientifica, IT-00133 Rome, Italy; demartino@bio.uniroma2.it (A.D.M.); beninati@bio.uniroma2.it (S.B.); 3Department of Biomedical Sciences and Surgical Specialties, University of Ferrara, Via Luigi Borsari 46, IT-44121 Ferrara, Italy; msc@unife.it; 4Department of Chemical and Pharmaceutical Sciences, University of Ferrara, Via Luigi Borsari 46, IT-44121 Ferrara, Italy; giordana.feriotto@unife.it

**Keywords:** tissue transglutaminase, cancer stem cells, inflammation, metastasis

## Abstract

Tissue transglutaminase (transglutaminase type 2; TG2) is the most ubiquitously expressed member of the transglutaminase family (EC 2.3.2.13) that catalyzes specific post-translational modifications of proteins through a calcium-dependent acyl-transfer reaction (transamidation). In addition, this enzyme displays multiple additional enzymatic activities, such as guanine nucleotide binding and hydrolysis, protein kinase, disulfide isomerase activities, and is involved in cell adhesion. Transglutaminase 2 has been reported as one of key enzymes that is involved in all stages of carcinogenesis; the molecular mechanisms of action and physiopathological effects depend on its expression or activities, cellular localization, and specific cancer model. Since it has been reported as both a potential tumor suppressor and a tumor-promoting factor, the role of this enzyme in cancer is still controversial. Indeed, TG2 overexpression has been frequently associated with cancer stem cells’ survival, inflammation, metastatic spread, and drug resistance. On the other hand, the use of inducers of TG2 transamidating activity seems to inhibit tumor cell plasticity and invasion. This review covers the extensive and rapidly growing field of the role of TG2 in cancer stem cells survival and epithelial–mesenchymal transition, apoptosis and differentiation, and formation of aggressive metastatic phenotypes.

## 1. Introduction

Despite the increased understanding of cancer cell biology and significant progression in surgery, cancer remains one of the leading causes of mortality worldwide. Although oncologists have a large number of therapeutic approaches to fight cancer, including chemotherapy, radiotherapy, immunotherapy, target therapy, and a combination of these, it is well demonstrated that resistance and recurrence phenomena often lead to a failure of conventional therapies [1,2,3].

Cancer develops when accumulation of genetic and epigenetic alterations brings a cell to autonomous growth through changes in its metabolism, a process that is supported by several mechanisms. Several studies underline the tight connection between inflammation and carcinogenesis [4]. Therefore, oxidative stress and chronic inflammation are essential in the initiation and promotion of cancer [5]. Inflammatory mediators, derived from cancerous, immunocompetent, and neighboring stromal cells, provide a supportive microenvironment for the neoplastic process. Cytokines such as interleukin (IL)-1β, IL-6, IL-8, tumor necrosis factor (TNF)-α, fibroblast growth factor (FGF), vascular endothelial growth factor (VEGF), and transforming growth factor (TGF)-β represent the major players in this intricate process [6] that also involves cancer stem cell (CSC) subpopulations [7]. These CSCs represent a small population of tumor cells resistant to conventional therapy, capable of self-renewal and differentiation, able to sustain long-term tumor growth, metastasis and tumor relapse. [8,9]. It is interesting to note that CSCs contribute to the cell heterogeneity of intra-tumor cells, in response to surrounding signals, selective pressure of anticancer therapies, and following the epithelial–mesenchymal transition (EMT) process [10,11], that includes phenotypic and metabolic alterations, such as loss of cell junctions, cytoskeleton reorganization, etc., leading to the promotion of metastatic characteristics in tumor cells [12].

Tissue transglutaminase (TG2) belongs to a family of enzymes (EC 2.3.2.13) that catalyze the post-translational modifications of proteins through a calcium-dependent acyl-transfer reaction between the γ-carboxamide group of a peptide-bound glutamine residue and the ε-amino group of a peptide-bound lysine (or other primary amine), leading to the formation of ε-(γ-glutamyl)lysine crosslinks [13,14]. This key multifunctional enzyme plays an essential role in CSC biology, and in the promotion and metastasis of tumors. Indeed, TG2 overexpression has been considered a negative prognostic marker, frequently associated with metastatic spread and drug resistance [15,16]. However, the role of TG2 in cancer is still controversial, since it has been reported as both a potential tumor suppressor [17] and a tumor-promoting factor [18]. Overall, it is clear that the functions of TG2 in cancer are cell type and situation specific, and generalizing the function of TG2 based on findings in a given cell type and circumstance often cannot be done. This review will provide a brief overview that considers the role of TG2 expression and activity in cancer initiation, survival, and progression.

## 2. Tissue Transglutaminase Description

The discovery of the catalytic domain of factor XIII in blood coagulation in vertebrates [19] pre-empted the subsequent discovery of a further eight members of the transglutaminase family. To date, several functional types of transglutaminases (TGs), which differ in their post-translational sequence and, possibly, structural specificity towards target proteins, have been characterized in human tissues [14,20]. These enzymes catalyze a variety of protein modifications associated with physiological and pathological conditions [21,22,23]. Conversely, a component of the family (human erythrocyte protein 4.2), is devoid of enzymatic activity and is thought to perform a purely structural role [24].

In contrast to the other members of its family, TG2 is a multifunctional enzyme (Figure 1) apparently involved in disparate biological processes [25,26]. Transglutaminase 2 catalyzes Ca^2+^-dependent post-translational modifications of proteins, generating protein–protein crosslinks, by a transamidation of γ-carboxamide groups of specific glutamine residues to form a thiol ester with the cystamine of the active site (releasing ammonia), and by a transfer of the acyl intermediate to a nucleophilic substrates such as the ε-amino group of lysine residues or biogenic amines [27]. Indeed, di- and polyamines, putrescine (PUT), spermidine (SPD), and spermine (SPM) are excellent substrates of TG2 [28]. The incorporation of these amines into proteins can occur through one or both of their primary amino groups, and result in the formation of either N-*mono*(γ-glutamyl)- or N,N-*bis*(γ-glutamyl)-PUT, -SPD, or -SPM. The crosslinked protein products are highly resistant to mechanical challenge and proteolytic degradation. For many years, TG2-mediated deamidation was considered to be a side reaction but, recently, substrate-specific deamidations have been reported [29].

Transglutaminase 2 is ubiquitously expressed both in intracellular and extracellular compartments where its localization is crucial for the regulation of its biological activities [30]. Due to its pleiotropic functions, TG2 activities are tightly regulated by Ca^2+^ ions, guanine nucleotides, and redox signals [14]. The interaction with Ca^2+^ or GTP induces conformational change that shifts TG2 in favor of transamidation activity or cell signaling functions [14]. Conversely, TG2 binding to GTP or GDP leads to a “closed” conformation that prevents its transamidation activity [31], while Ca^2+^ binding to TG2 leads to an “open” conformation that does not allow GTP/GDP binding, keeping the active site available for the protein crosslinking reaction. Generally, TG2 catalytic activation requires high (>1 mM) Ca^2+^ concentrations. Therefore, low Ca^2+^ concentration and high GDP/GTP levels in cytosol can prevent crosslinked activation. Nevertheless, a large variety of TG2 crosslinking cellular substrates have been identified [32]. Hence, in order to produce local intracellular specificity of TG2 activation, Ca^2+^ levels could increase locally, for example, as consequence of its release from lysosomes in autophagic processes [33,34]. Interestingly, some molecules could modify the TG2 Ca^2+^ requirement. Indeed, it has been demonstrated that phingosyl phosphocholine, a membrane component, could serve as a cofactor and bind TG2 in order to enhance the affinity of the enzyme for Ca^2+^ [35]. It is tempting to speculate, therefore, that different conformations determine opposite effects on cancer cell fate [36,37]. Transglutaminase 2 closed form is considered a pro-survival factor for cancer and CSCs [38,39]. Conversely, the open conformation may be considered, at least in some contexts, a tumor suppressor factor [17,37].

The expression of TG2 is tightly controlled and can be induced at the transcriptional level by cytokines and the activation of various transcription factors [40,41]. Moreover, the alternative splicing of TG2 gene (*TGM2*) mRNA has been observed [42,43]. The constitutively spliced full-length transcript (TGM2_v1), which encodes a polypeptide of 687 amino acids (with molecular weight of ~75 kDa) was first reported by Gentile and colleagues [44]; four additional transcripts have subsequently been described. In human erythroleukemia cells, ~62 kDa (TGH or TG2-S or TGM2_v2) and ~38 kDa (TGH2 or TGM2_v3) N-terminal fragments have been identified [45]. In these isoforms, the loss of C-terminal GTP binding domain, which is also responsible for regulation of TG2 Ca^2+^-dependent activation, represents the main trait of these variants. The so-called short isoform (TGH) has been related to neurodegenerative disorders [46] and to neuroblastoma cell differentiation [36]. Recent data suggests that increased expression of the TGH2 variant may contribute to complex celiac disease physiopathology [47]. Two additional C-terminal truncated isoforms (tTGv1 or TGM2_v4a and tTGv2 or TGM2_v4b) have been identified in human aortic vascular smooth muscle cells and leukocytes [48]. Interestingly, it has been recently demonstrated that these variants are predominantly expressed in peripheral blood mononuclear cell (PBMC)-derived primary progressive multiple sclerosis patients with respect to full-length TG2 [49]. Although the role of alternative spliced isoforms of TG2 remain unclear, growing evidence indicates that splicing variants can provide diagnostic and/or therapeutic targets for several cancers [50], opening new perspectives in this field.

Transglutaminase 2 exerts, also, a protein disulfide isomerase activity (PDI), involved in the formation and breakage of disulfide bonds between cysteine residues [51] and, probably, with regulation of mitochondrial functions and apoptosis [52,53,54]. Recently, it has been reported that TG2 PDI activity is responsible for activation of heat-shock factor 1 (HSF1), leading to regulation of proteostasis [55]. Transglutaminase 2 intrinsic serine/threonine kinase activity has been also reported [56,57]. Transglutaminase 2 plays a key role in the promotion of cell adhesion, forming a complex with fibronectin in a transamidation-independent manner [58].

Due to its several enzymatic and non-enzymatic functions [59,60], TG2 regulates several physiological processes, such as apoptosis, differentiation, inflammation, and fibrogenic reactions [61,62], via post-translational modifications of a large variety of substrates [32]. Thus, alteration of TG2 activity and/or its regulation causes many types of diseases, such as neurodegenerative, autoimmune, inflammation-related diseases, and cancer [63,64,65,66].

## 3. Interplay between Tissue Transglutaminase and Cytokines in Inflammation and Cancer Progression

The close relationship between inflammation and TG2 has been variously demonstrated in several physiological and pathological conditions [64,67,68,69], including cancer [40,70]. Transglutaminase 2 expression and transamidation activity are often increased in inflammatory events, and several cytokines and growth factors are able to regulate TG2 expression and activity. In particular, *TGM2* is regulated by canonical and non-canonical nuclear factor (NF)-κB pathway [71,72], that represents a key player in inflammation, cancer development, and progression [73]. For example, TNF-α, IL-1, and IL-6 are the principal target genes of NF-κB and, at the same time, these cytokines are known to be potent inducers of TG2 expression [40].

A recent report demonstrated that transforming growth factor-β (TGF-β), at least in the advanced stages, represents a key promoter of cancer progression and metastasis [74], and it is able to induce cancer EMT [75]. Furthermore, TG2 expression and TGF-β-induced EMT are closely related [76,77,78,79]. Interestingly, it has been suggested that extracellular TG2 crosslinking activity or TG2 expression are required for angiogenesis, through a mechanism that involves matrix-bound VEGF [80] but also matrix-bound TGF-β [81]. Therefore, the role of TG2 crosslinking activity in the regulation of cancer progression or inflammation represents an important field of study. For example, angiogenesis is a well-regulated and essential process for cancer development, in which the principal players are members of the VEGF family and their receptors [82]. Moreover, during cancer progression, this phenomenon is regulated also by platelet-derived growth factor (PDGF), FGF, and TGF-β [83,84,85]. It is interesting to note that PDGF-BB also shows an essential chemotactic activity in several tumors [86,87]. It has been reported that PDGF-BB is a TG2 substrate, leading to the formation of high-molecular weight complexes that impair glioblastoma cell migration [88]. This indicates that TG2 crosslinking activity may modulate the functions of growth factors, or other extracellular signaling molecules, through the formation of complexes with possible altered biological activity.

Recently, Willis and colleagues [89] have demonstrated that TG2 catalyzes the formation of high-mobility group box 1 (HMGB1) complexes with several proteins, including autoantigens. Since nuclear localization of TG2 has relevant physiological and pathological implications [90], the authors propose that TG2-HMGB1 interactions modify TG2 nuclear localization, thus regulating gene expression. High-mobility group box 1 plays a pleiotropic role in inflammation and cancer progression [91] and, hence, the interplay between TG2 and HMGB1 could be important for future applications in the field of cancer progression.

## 4. Role of Tissue Transglutaminase in Cancer Stem Cell Biology

At least for some types of cancer, progression of primary tumors to metastatic disease is sustained by CSCs (or tumor-initiating cells, TICs) that represent a subpopulation of cancer cells with the ability to survive and promote tumor growth [9,92]. Several authors have proposed that TG2 expression increases the cellular plasticity and changes the dynamic equilibrium from non-stem cells towards stem cell compartment. Illopoulos and colleagues [93] arrived at a similar conclusion, supporting the notion that tumor heterogeneity involves a dynamic equilibrium between stem and non-stem cell compartments, and that this plasticity is regulated by IL-6. Evidence that TG2 expression promotes an increase in stem cell-like properties, suggesting novel strategies for targeting CSCs/TICs subpopulations in tumors for improved clinical outcomes. The most well-recognized CSC molecular marker is CD44, a non-kinase transmembrane glycoprotein that is expressed on embryonic stem cells and is able to bind several ligands, including hyaluronic acid. CD44 plays a critical role in tumor progression, chemoresistance, and metastasis through the activation of several cell signaling pathways [94]. Transglutaminase 2 expression is linked to an increase in CD44^+^ (and CD24^−^) subpopulations in breast cancer cells, leading to promotion of stem-like properties and a metastatic phenotype [76,95]. Interestingly, these authors suggest that TG2 may not only induce EMT in the primary tumor but, also, confer stemness-associated properties, including plasticity and self-renewal ability in secondary tumors, thus supporting the formation and growth of metastases [95]. This observation is supported by recent data demonstrating that inhibition of TG2 expression leads to the disappearance of CSC surface antigens (including CD44) in renal cell carcinoma [96]. In fact, in this cancer model, TG2 expression silencing compromises cellular adhesion, migration, and invasion by blocking the binding on β1-integrin, type I collagen, and laminin. By contrast, other data indicate that loss of the enzymatic and adhesive activities of TG2 impairs cell–matrix interactions, leading to the decrease of adhesion and promotion of cellular motility [97]. Therefore, it appears that the role of TG2 activity in these phenomena remains strongly dependent on its cellular context.

Fu and colleagues demonstrated that TG2 inhibition with monodansylcadaverine (MDC) suppresses cell proliferation and induces apoptosis in CD44^+^ glioma stem cells through the decrease of inhibitor of DNA binding 1 protein (ID1) and that this, therefore, may be a TG2 downstream regulator [98]. In this study, the use of MDC in the treatment of CD44^+^ glioma cells has been extended in vivo. The role of transamidation activity in CSC survival has been also underlined in epidermal cancer stem cells [99]. In this cellular model, the use of NC9, an irreversible TG2 inhibitor, locks the enzyme in the open conformation and reduces the expression of EMT-related proteins. Surprisingly, the presence of GTP-binding functions is necessary for the survival and EMT phenotype [99]. Therefore, the site-specific inhibition of transamidation activity is also able to reduce CSC growth through a protein structure conformational change that results, however, in the inactivation of the TG2 GTP-binding site [100]. Similar results were found in mesothelioma CSCs [101]. In these cells, TG2 expression is essential to sustain spheroid formation, migration, and invasion, and the use of NC9 inhibitor impairs EMT phenotype and metastatic potential. Transglutaminase 2 expression is markedly increased in human colorectal CSCs, and knockdown of TG2 by specific RNA interference clearly inhibits cell growth, stemness, and metastatic ability of these cells [102]. The role of TG2 in ovarian cancer EMT is also well established. The Matei group [103,104] clearly demonstrated that activation of oncogenic signaling by TG2 induces EMT and enhances invasiveness and ovarian tumor dissemination. Epithelial–mesenchymal transition also seems independent of the TG2 transamidation activity in mammary epithelial cells [105]. Yakubov and co-workers [106] have demonstrated that the principal pathway activated by extracellular TG2 is the non-canonical NF-κB signaling, which leads to the upregulation of CD44 and promotes EMT phenotype. More recently, it was reported that the functional inhibition of TG2 fibronectin-binding domain suppresses the formation of complex between TG2, fibronectin, and β1-integrin, and reduces spheroid formation [107]; in this case, TG2 promotes ovarian CSCs survival also through the Wnt/β-catenin pathway, by direct interaction with Wnt receptor Frizzled 7 (Fzd7), that is generally upregulated in ovarian cancer with respect to normal ovarian epithelium [107]. Also in this case, promotion of tumor-initiating capability is independent of TG2 transamidation activity.

Recent data suggest that TG2 plays a significant role in recurrence and radioresistance of glioblastoma multiforme, which is considered the most malignant glial tumor, with very poor survival. In fact, irradiation induces TG2 upregulation in necrotic areas surrounding the tumor, which is responsible for the induction of mesenchymal transdifferentiation of glioma stem cells (GSCs) with subsequent radioresistance development. This process is supported by secretion of macrophages/microglia-derived cytokines (e.g., TNF-α) [108]. Interestingly, TG2 has different expression levels in the various GSCs subtypes. It has been demonstrated that TG2 is specifically expressed in more aggressive mesenchymal (CD44^+^) GSCs, but not in less aggressive proneural GSC subtype, in which TG2 can be induced by aldehyde dehydrogenase 1A3 (a retinaldehyde dehydrogenase) and retinoic acid, opening new possible strategies to fight this type of cancer [109]. Moreover, TG2 seems downregulated in temozolomide-resistant GSCs, together with the reduction of mesenchymal features and sensitivity to radiation [110].

The close connection between TG2 and inflammation is also reflected in the CSC model. Pro-survival activity of TG2 is mediated by TGF-β production in several tumor types. Indeed, in ovarian cancers cells, TGF-β induces TG2 expression with consequent stimulation of EMT and cancer stem cell phenotype (CD44^+^) [77]. Similar studies on colorectal [79] and breast cancer cells [111] corroborate these findings. Another important pro-inflammatory cytokine IL-6, present in tumor microenvironments, is involved in the EMT and in the acquisition of a stem-like phenotype of cancer cells [112]. Oh and colleagues [113] have clearly demonstrated that IL-6 production is strictly dependent on TG2 expression which, in turn, mediates the in vitro tumor sphere formation of ovarian cancer cells. Moreover, the authors underline how TG2/IL-6 axis is able to promote CSCs, EMT, and metastatic phenotypes; in breast cancer, this mechanism is increased by IL-1β production [114].

## 5. Tissue Transglutaminase in Cell Death and Differentiation

Several lines of evidence underline the opposing roles of TG2 in cell death and survival [62,115,116]. The pro-apoptotic effect of TG2 is based on its calcium-dependent crosslinking activity [117]. In fact, elevation of intracellular calcium levels, due to stressful conditions and consequent release from endoplasmic reticulum (ER), leads to the induction of TG2 transamidation activity. This activity catalyzes the formation of apoptotic bodies around dying cells [118] and induces cell death through post-translational modifications of pro-apoptotic proteins [119]. Conversely, TG2 anti-apoptotic activity is generally independent from its crosslinking activity. Indeed, this pro-survival signal is principally mediated by adhesive capabilities of TG2 and integrins to fibronectin, which impairs anoikis (detachment-induced apoptosis) [120,121]. In addition, it has also been reported that TG2 can bind to cathepsin D via crosslinking, leading to apoptosis impairment [122]. Therefore, the mechanisms by which TG2 prevents apoptosis are dependent on cellular localization and on the activation of specific pathways [18,123], including inflammatory ones (e.g., NF-κB pathway) [71,124]. The tight association between TG2, inflammation, and cell survival is highlighted, for example, by the complex role of TGF-β [67], a pleiotropic molecule that may act as a tumor promoter or tumor suppressor, depending on stages of carcinogenesis [125]. Indeed, TG2 is required for the activation of TGF-β [126], and this cytokine is necessary for TG2 expression and activity [77]. In addition, it has been demonstrated that in TG2 knockout (KO) mice, TGF-β levels are decreased, as well as the macrophage capability to engulf apoptotic bodies [127].

Polyamines are naturally occurring and ubiquitous polycationic alkylamines that represent positive and essential regulators of cell growth and proliferation. Although high polyamine levels have been associated with cancer progression [128], it has been demonstrated that TG2 activity is augmented in polyamine-induced apoptosis, underlining another pathway involved in TG2-mediated cell proliferation [129]. Interestingly, polyamines (i.e., SPD) are able to induce autophagy (a self-degradative process essential for removing misfolded or aggregated proteins and clearing damaged organelles in response to stress) in human cells [130,131]. Therefore, the interplay between polyamines and TG2 in autophagy represents an intriguing research field [132]. Indeed, it is well-known that TG2 promotes autophagy [133], as supported by the observations that, in TG2 KO mice, autophagosome maturation is impaired [134,135].

The role of TG2 in apoptosis and autophagy results from other regulated pathways. In renal cell carcinoma, TG2 expression promotes cell survival through crosslinking of p53 in autophagosomes and subsequent p53 depletion [133,136]. Interestingly, p53 is also a substrate for TG2 serine/threonine kinase activity [137], suggesting further mechanisms by which TG2 could facilitate apoptosis. Since the most lethal p53-mutant tumors possess a low grade of differentiation, differentiation-restoring treatments represent an alternative therapeutic strategy [138]. Due to its low toxicity profile, differentiation therapy may be employed in some cellular model of cancer [139,140]. It is well-known that all-*trans* retinoic acid (ATRA) represents the prototype of differentiation therapy agents which have the ability to arrest cell growth, inducing differentiation and apoptosis in promyelocytic leukemia cells [141,142]. This phenomenon is accompanied by an increase in TG2 expression and transamidation activity [143,144,145]. Moreover, it has been demonstrated that, in some cancer cellular models, most differentiation inducers are able to increase TG2 crosslinking activity, leading to reduction of proliferation and metastatic potential [146,147,148,149]. These observations underline the tumor suppressor activity of TG2.

## 6. Role of Tissue Transglutaminase in Cancer Metastatic Cascade

The transition from benign to malignant phenotype is the key event for the metastatic progression of tumors. The metastatic cascade is characterized by several distinct phases, such as degradation of basement membrane, movement towards the blood or lymph vasculature, attachment to endothelium, extravasation, adhesion to extracellular matrix (ECM), migration, and invasion [17]. The metastatic process is due to changes in cellular plasticity, including EMT, amoeboid transition, and mesenchymal-to-amoeboid transition. These morphological and cytoskeletal mutations of tumor cells are necessary to facilitate cell locomotion and create the basis for ECM remodeling [150]. Transglutaminase 2 has been found to be essential for all these steps of metastatic processes [17,151].

The interactions between cells and the ECM represent a quite complex phenomenon which provides stromal and tumor cells with mechanical support for adhesion. One of the most documented functions of TG2 in cancer spreading is their involvement in tumor cell–ECM interaction [58]. Transglutaminase 2 present in the ECM supports the adhesion of cancer cells to the matrix, and cell surface TG2 is associated with integrins and mediates cell–matrix adhesion. The integrins are important proteins responsible for the regulation of adhesion, migration, and invasion, having, as a substrate, fibronectin, collagen, and laminin [152]. Inhibition of integrin interaction with ECM proteins leads to apoptosis, which can be prevented by direct interaction of the TG2 with fibronectin in the outer cellular membrane with the fibronectin [153]. Generally, the involvement of TG2 in cell–matrix interactions does not derive from transamidation [58,154]. The observation that TG2 overexpression is responsible for tumor cell adhesion and motility, whereas its reduction impairs these functions [155,156], corroborate these findings. Nevertheless, several communications disprove these observations. In fact, it has been hypothesized that TG2 favors cell adhesion by the changes induced in the ECM by means of protein crosslinking [120]. There is also reliable evidence that post-translational modification of ECM proteins by TG2 is a key step for the progression of tumor cells to metastasis, conferring resistance to metalloproteinases and promoting cell–matrix interactions, particularly via crosslinking of fibronectin and collagen [96,157].

The involvement of TG2 in cancer cell motility and invasion appears controversial, and different effects have been observed after protein overexpression or enhancement of the enzyme activity through specific activators [17]. Moreover, the role of TG2 in cancer cells migration depends on tumor staging. Indeed, at the early stages of tumor progression, the decrease of cell surface TG2 induces cancer cells migration [97]. Conversely, TG2 upregulation has been associated with the late-stage aggressive metastatic cancer phenotype and, possibly, with an increase of drug resistance [104,158,159]. The downregulation of endogenous TG2 by siRNA inhibits fibronectin-mediated cell adhesion and invasion, whereas TG2 overexpression improves these functions [158,160]. Based on the overall reported, in order to improve knowledge on the role of TG2 in tumor progression, it can be concluded that it is necessary to distinguish the effects produced by protein expression from those produced by its transamidation activity, since both of these essential functions participate differently in various phases of invasion and tumor growth.

Many TG2 substrates, including tubulin, actin, vimentin, myosin, laminin, and spectrin, are involved in cytoskeleton organization [32], and it is well established that TG2 co-localizes with stress fibers [161]. In fact, cytoskeleton plasticity is integral to cell migration and other metastatic processes. A proteomic study on lamin A knockdown in a colorectal cancer cell line emphasizes the role of TG2 in differential post translational modifications of other cytoskeleton proteins, in a mechanism that impairs the formation of dynamic actin filaments over stress fibers, accounting for the altered cell motility properties [162]. The regulation of cytoskeleton rearrangement, as well as cellular growth and differentiation, also involves the members of Rho family [163]. Several studies underscore that TG2 directly activates RhoA via both transamidation and non-enzymatic activities [164,165]. Moreover, as an alternative mechanism in the modulation of cellular plasticity, TG2 is able to bind Bcr protein, a regulator of the Rho family [166].

Transglutaminase 2 plays an important role in ECM stiffness and can regulate the cell contractile response by modulating actin polymerization and myosin light chain phosphorylation [167]. However, it has been demonstrated that plasticity and metastatic potential of cancer cells can also be affected by the degree of TG2-mediated post-translational modification of proteins with polyamines [168]. In fact, the presence of protein-bound polyamines is inversely correlated to metastatic potential (Figure 2). Moreover, the TG2-mediated post-translational modification of polyamines of laminin or Matrigel (a reconstituted basement membrane) impairs the adhesiveness of a melanoma cell line [169], and evidence showing that TG2 overexpression and crosslinking activity have protective effects against in vivo melanoma progression corroborate these findings [170]. On the contrary, the absence of TG2 represents a favorable condition for metastasis formation [171]. This seemingly contrasting role of TG2 in metastasis might therefore depend on the local availability of its protein substrate(s) [170].

## 7. Tissue Transglutaminase and Angiogenesis

Angiogenesis, the process of blood vessel growth, includes the proliferation, migration, and differentiation of endothelial cells into tubular structures. Angiogenesis is essential in tumor growth, since it enhances nutrients and oxygen supply, and plays a crucial role for distant metastasis dissemination. The tumor microenvironment is rich of pro-angiogenic factors produced by neoplastic, stromal, and infiltrating immune cells [172], such as VEGF, FGF, and TGF-β. Transglutaminase 2 is highly expressed in endothelial cells, with positive effects on tubule formation [173,174]. In particular, it has been demonstrated that inhibition of TG2 expression and transamidation activity leads to inhibition of angiogenesis [80], via regulating the deposition of VEGF into the ECM and, in turn, facilitating activation of its signaling through VEGF receptor 2 (VEGFR2). In addition, the inhibition of GTP-binding activity of TG2 also suppresses angiogenesis [175]. However, the role of TG2 in angiogenesis remains contradictory. In fact, it has been suggested that TG2 crosslinking activity may play an active role in wound healing and angiogenesis [176].

## 8. Conclusions

Transglutaminase 2 has been implicated in a large variety of cellular functions, many of which are crucial in tumor cell proliferation, survival, and metastatic spread. However, due to its pleiotropic activities, the role of TG2 in tumors is still controversial. In this context, it is well established that TG2 has opposing role in apoptosis, depending on intracellular calcium levels or on the GTP-binding activity. Transglutaminase 2 functions are also related to the dissemination of tumor cells, due also to its role as a cell surface receptor, which can mediate tumor cell–ECM interactions, or modulate tumor angiogenesis. Moreover, several recent studies implicate TG2 as a regulator of EMT and underline why its expression is markedly increased in CSCs.

Based on the literature data, it is reasonable to hypothesize that TG2 selectively affects tumor-associated events, including tumor-related inflammation, by modulating its enzymatic and non-enzymatic properties in relation to the cell type, its localization within the cell, local substrate availability, and intra- and extracellular environments. In conclusion, TG2 appears to be more increasingly involved in cancer biology and, so, development of selective inhibitors and translational studies (also on natural inducers) seem essential for gaining a more precise and complete view of the role of TG2 in cancer development and progression.

## Figures and Tables

**Figure 1 medsci-07-00019-f001:**
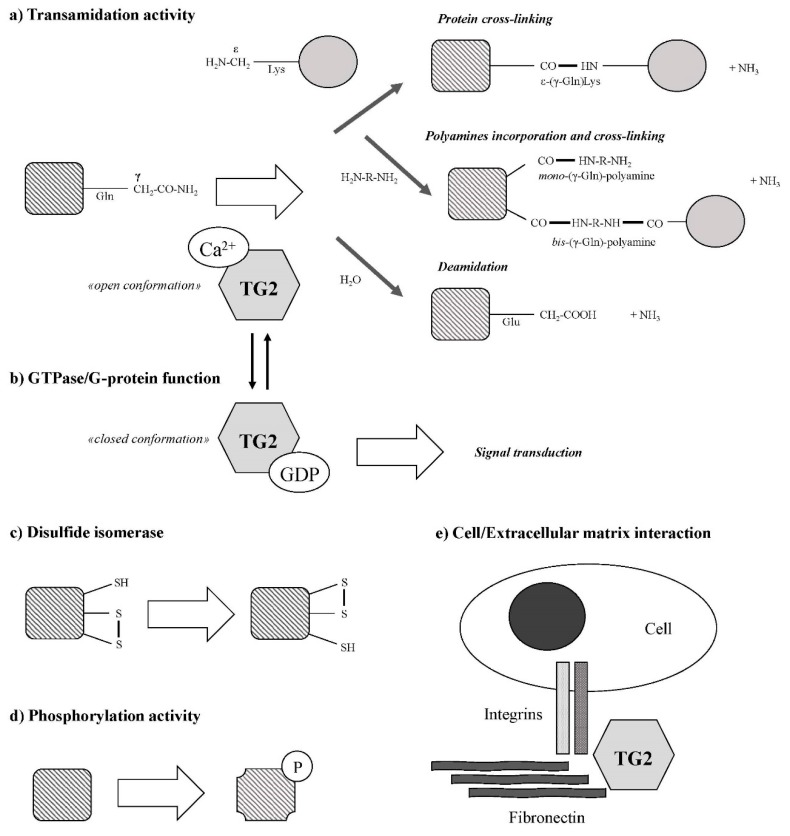
Reactions catalyzed by type II transglutaminase or tissue transglutaminase (TG2) (from Tatsukawa et al. [62] with modifications).

**Figure 2 medsci-07-00019-f002:**
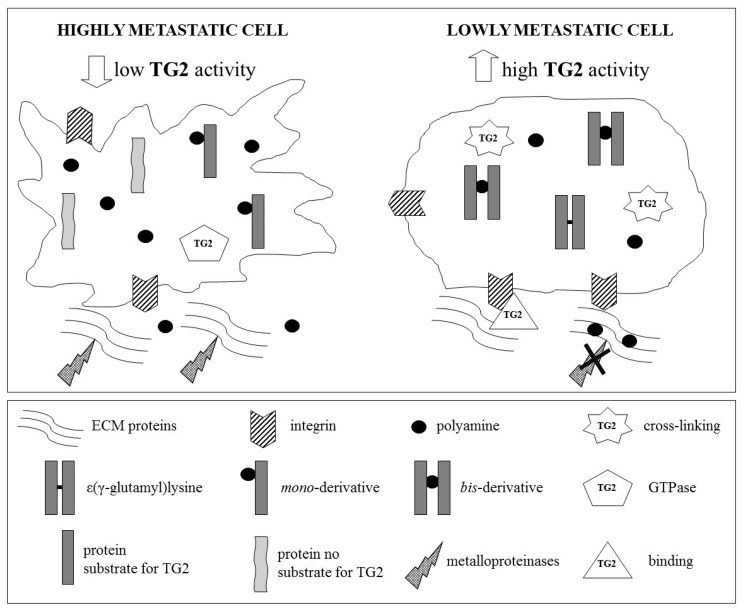
Tumor cells, with high metastatic potential, are characterized by high levels of intracellular polyamines and low transamidase activity of TG2. This condition favors the formation of mono-(γ-glutamyl) derivatives of polyamines, preventing polymerization of intracellular proteins, in particular, those components of the cytoskeleton. It follows that cellular plasticity is high, due to the reduction of cell–cell or cell–matrix interactions, which favors cell detachment and motility. By contrast, in cells with reduced metastaticity, the transamidase activity of TG2 is higher. Hence, the polymerization of intracellular proteins is increased by the favored formation of ε-(γ-glutamyl) lysine and bis-(γ-glutamyl) polyamine derivatives. Therefore, in these cells, the plasticity is reduced and the cell–matrix interactions, mediated by integrins and TG2, are more pronounced, due to the increase in cell adhesion with extracellular matrix (ECM) (modified from Lentini et al., 2013 [17]).

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
