# Peer review of "The Role of Tissue Transglutaminase in Cancer Cell Initiation, Survival and Progression"

_medsci, 2019, doi:10.3390/medsci7020019_

Round 1
Reviewer 1 Report
The manuscript “ The role of tissue transglutaminase in cancer cell initiation, survival and progression” was aimed to describe TG2 involvement in all stages of carcinogenesis, with different effects depending on its expression or activities, cellular localization and specific cancer model.
This paper can be of relevance in the field, to give evidence for different TG2 role. The manuscript is well written and should be of great interest to the readers.
However, some criticism persists
Although TG2 is a calcium-dependent enzyme, overexpressed in various cancer cells, few comments have been reported to emphasize different role of TG2 and inhibition of enzyme activity associated to calcium availability. Some sentence should be added.
Author Response
We thank the reviewers for their careful reviews and very helpful comments.
“Although TG2 is a calcium-dependent enzyme, overexpressed in various cancer cells, few comments have been reported to emphasize different role of TG2 and inhibition of enzyme activity associated to calcium availability. Some sentence should be added”.
Response: Some sentences have been added (red color) in section 2.
Reviewer 2 Report
I think that this subject is emerging and that there is room for reviews in this area. The manuscript does contain relevant material and there is an attempt to draw strands together but I don't feel it is complete and other reviews deal with the wide-ranging pleiotropy perfectly well and so some of the text could be lost without overall lack of quality.
My main issue is a lack of integration that made the reading of the semi-repeats in different sections less pleasurable. The continual "flagging up" of "as mentioned earlier" is an example of this. There are quite a lot of typos and some sentences need rewriting.
Paragraph 2 of section 4 and paras 2, and 3 of section 6 contained material that I was interested in but simply found the narrative too convoluted and difficult to follow.
Probably, the angiogenesis should be considered separately in section 7.
More on splicing needed.
Collectively, I think there is something worth continuing to work with here but I don't consider it ready for publication at this stage. If it can be drawn together more in a subsequent version that clarifies further the role of TG2 in generating or inhibiting cellular plasticity (parts of which are starting to come together here) and how this may relate to cancer and metastasis then I can see the point of this as a novel review. At this stage though still needs a bit of work.
Author Response
We thank the reviewers for their careful reviews and very helpful comments.
“My main issue is a lack of integration that made the reading of the semi-repeats in different sections less pleasurable. The continual "flagging up" of "as mentioned earlier" is an example of this. There are quite a lot of typos and some sentences need rewriting”
Response: The manuscript has been revised as indicated.
“Paragraph 2 of section 4 and paras 2, and 3 of section 6 contained material that I was interested in but simply found the narrative too convoluted and difficult to follow”
Response: These paragraphs have been modified (red color).
“Probably, the angiogenesis should be considered separately in section 7.”
Response: Angiogenesis has been considered as new section.
“More on splicing needed”
Response: Some sentences have been added (red color) in section 2.
“Collectively, I think there is something worth continuing to work with here but I don't consider it ready for publication at this stage. If it can be drawn together more in a subsequent version that clarifies further the role of TG2 in generating or inhibiting cellular plasticity (parts of which are starting to come together here) and how this may relate to cancer and metastasis then I can see the point of this as a novel review. At this stage though still needs a bit of work”
Response: Some sentences have been added in section 4 and in section 6.
Round 2
Reviewer 2 Report
A much improved review that draws together some interesting points regarding TG2 and its numerous roles that relate to human cancer. I am interested in this area having worked in the field on and off for nearly thirty years and feel that the subject itself is starting to come together and that this review helps push along the story.
Edit points
Line 22... remove "a"
Line 24... "hydrolysis" not hydrolyzing
Line 25...rewrite/ replace text so it runs "protein kinase, disulphide isomerase activities and is involved with cell division"
Line 26 ...remove "one of" replace with "a"
Line 26 ...remove "may be" replace with "is"
Line 26... remove ", even though" and replace with semi-colon ";"
Line 30.. replace "to" with "with"
Line 31....I don't think "contrast" is the correct term here - not sure what the sentence is trying to say
Line 34....remove "and" and plural phenotypes not "phenotype"
Line 45....remove "like" replace with "including"
Line 49...remove "and" replace with "through"
Line 49 ...replace with "a process that is supported by multiple mechanisms."
Line 51....carcinogenesis
Line 52 .... replace with "cancerous" and "supportive"
Line 55.... "that also involves cancer stem cells"
Line 56... subpopulations (plural)
Line 56... Replace "Indeed" with "These"
Line 60... consider ..."(EMT) process [10-11] that include phenotypic and metabolic alterations such as loss of
Line 62... New paragraph
Line 66....Consider ..."This key multifunctional enzyme plays an essential role"
Line 72... Remove "about" consider ...."that considers the role of TG2 expression"
Line 77...consider "in vertebrates ][19] pre-emptied subsequent discovery of a further (insert number) intracellular members of the transglutaminase protein family."
Line 79...consider "which differ in their post-translational sequence and possibly structural specificity towards target proteins which have been characterised in human tissues"
Line 80...consider "these enzynes catalyse a variety of"
Line 82... consider "devoid of enzymic activity and is thought to perform a purely structural role"
Line 83... remove "this protein" replace with "its"
Line 84 remove "structurally and functionally related enzyme" this is largely redundant as all enzymes have these features and rewrite
Line 86.. rewrite so that it does not sound as if deamination occurs before amine attack - the events are concurrent.
Line 87 consider..."biogenic amines [27] such as"
Line 89..... consider ""groups and result in the formation"
Line 93...Try not to start too many sentences with "TG2" as it get repetitive
Line 93... remove "and is present"
Line 93... consider "expressedin both intracellular and extracellular compartments where its localalisation"
Line 97 remove "Indeed" replace with "Conversely,"
Line 102 "cross-linked" not "cross-linking"
Line 102 ...consider ..."In order to produce local intracellular specificity of TG2 activation"
Line 106...consider inserting .... "It is tempting to speculate therefore that different conformations"
Line 110.... see Line 93
Line 110.... remove "along with" replace with "and"
Line 111. ....remove "an" and insert "the" before TG2
Line 114....insert semi-colon after [44] and remove "Besides this transcript"
Line 114 ...consider "have subsequently been described"
Line 119... "suggests" not "suggest"
Line 159/160... "This indicates"
Line 162..".catalyses" not "catalyse"
Line 169.... cell not "cells"
Line 173..."Several authors have proposed that TG" expression increases"
Line 174 ..."shifts" not "shift" (this verb is slightly colloquial and used too much throughout the review"
Line 175.... "arrived at a similar conclusion supporting the notion that"
Line 176 ...."and that this plasticity"
Line 177... delete text and rewrite ""promotes an increase in stem cell like properties suggesting novel strategies"
Line 179.... replace the most" by "best"
Line 179 "CSC" not "CSCs"
Line 180...replace with "transmembrane glycoprotein that is expressed"
Line 180 "and is able to"
Line 182...remove comma
Line 184...."and a metastatic phenotype"
Line 186... remove first comma
Line 190..".indicates" not "indicate"
Line 192...consider "Therefore, it appears that the role of TG2 activity in these phenomena remains strongly dependent on its cellular context"
Line 196.....remove "which" replace with "and that this therefore may be"
Line 200... "binding functions" not "bounding function"
Line 207 ..."In fact" comes up too often...consider alternative tem
Line 211.....leads not lead
Line 212..."promotes" not "promote"
Line 216..consider ".[107]; in this case"
Line 230..."mediated by" not "mediate by"
Line 232... move [111] to after "cancer cells"
Line 234...replace "to" with "of"
Line 237..add semi-colon after "phenotypes" and remove "the"
Line 241...consider "Several lines of evidence underlie the opposing"
Line 250... consider another term rather than "cellular contest"
Line 258... polyamine not polyamines
Line 265... change to "autophagosome maturation is impaired"
Line 270...consider "alternative therapeutic strategy"
Line 275...consider "cellular models of cancer"
Line 276....comma after "activity"
Line 288...insert "these" before "steps"
Line 295...insert comma after "invasion"
Line 296...consider "interaction of TG2 with fibronectin in the
Line 298....Remove "Data suggesting that" and replace with "The observation that"
Line 302...remove "are currently" replace with "is also"
Line 302..."evidence" not "evidences"
Line 305...remove "in fact" and replace with "and "
Line 307 - see line 207
Line 308...remove "the"
Line 309...replace with ."(and possibly with"
Line 310....remove Data suggesting that" and replace with "The"
Line 310 ... remove comma after RNA
Line 311 ....inhibits not inhibit
Line 311.. improves these functions"
Line 312....remove "Based on...........tumor progression. sentence
Line 313 ...start new sentence...It can be
Line314.....comma after "activity" and continue sentence ...."since both these....."
Line 318....remove "the"
Line 322...edit "also involves" not "involves also"
Line 326...."plays" not "play"
Line 328 "can also be affected"
Line 331.... brackets around "a reconstituted basement membrane"
Line 332... remove full stop after [169] insert edit "and evidence showing"
Line 335.....consider....might therefore depend on the local availability of its protein substrate(s)"
Line 339.... might benefit from a sentence describing the importance of the role of angiogenesis in cancer at the start of the paragraph.
Line 340.... remove "it"
Line 342 remove "was" replace with "is"
Line343... formation not formations
Line 350...Conclusion heading on new page
Line 352....start the paragraph with the second sentence then start the next sentence with "However, due to....."
Line 359...."TG2 selectively affects tumor-associated events"
Line 361....insert "its" before localisation
Line 361 try..."local substrate availability"
Line 362....remove full stop and try...and so development of selective inhibitors ........(also on natural inducers) seem essential for gaining a more precise"
I haven't checked the references - ran out of time - but might be worth a final double check through
"
Author Response
We thank the reviewer for his careful reviews and very helpful comments.
The manuscript has been improved (blue color) according to reviewer’s suggestions.
